# VARIATIONAL AGENT DISCOVERY

## ABSTRACT

We introduce Variational Agent Discovery (VAD), an unsupervised agent representation learning algorithm that discovers agent-centric representations directly from pixels. We frame agent representation learning as a prediction problem where we aim to predict what latent actions describe transitions of latent variables used to model a scene. VAD leverages slot-based attention with a variational objective that jointly learns inverse dynamics (inferring actions from transitions), forward dynamics (predicting states from actions), and agent policies (distributions over actions). Without any supervision, VAD develops representations that generalize to novel agents and goals with minimal performance degradation. Our learned representations enable downstream tasks like action prediction and goal inference. Notably, VAD exhibits shared action representations across multiple observed agents—feature dimensions that consistently activate for the same action regardless of which agent performs it—and demonstrates teleological reasoning capabilities similar to 12-month-old infants, suggesting that these cognitive phenomena can emerge from our unsupervised agent representation learning objective.

## 1 INTRODUCTION

Unsupervised learning of agent-oriented representations is a fundamental unsupervised learning challenge. While object-centric learning methods can successfully decompose scenes into entity representations, they do not explicitly model the distinction between entities that act with purpose versus those following environmental dynamics—a distinction that may be critical for tasks like action prediction, goal inference, and adaptation to novel agents. Even from minimal visual cues like geometric shapes moving on a screen (Heider & Simmel, 1944), humans can discern agency and goal-directed behavior (Csibra, 2008; Csibra et al., 1999), making this information available for prediction, reasoning, and planning (e.g., Shu et al., 2020). This suggests that it is possible to learn agent-oriented representations from visual observation alone. Accomplishing this may be a key step in developing artificial agents with human-level social reasoning abilities.

Most models of how humans accomplish agent perception operate on preprocessed scene representations, not on pixels. Thus, a significant aspect of the computational problem is sidestepped in these models. Consider the challenge of learning from tutorial videos or gameplay demonstrations. The observer sees only pixels—no labels indicate which entities act with purpose versus those following environmental dynamics. Even state-of-the-art vision-language models struggle with this challenge: Schulze Buschoff et al. (2025) found that multimodal LLMs fail to match human judgments in visual agency tasks. A similar limitation applies to current imitation learning from observation (ILFO) methods (Torabi et al., 2018a;b; Edwards et al., 2019; Young et al., 2021; Lee et al., 2021; Choi et al., 2023; Huang et al., 2024; Zhou et al., 2023), which either require pre-identified agent trajectories or operate in single-agent settings where agency is assumed.

We address this challenge by introducing *Variational Agent Discovery* (VAD), an unsupervised approach that discovers agent-oriented representations and learns their behavioral models simultaneously. VAD leverages a factored neural network composed of multiple state factors or "slots" that each learn an attention mechanism over visual features representing a scene. Our key insight is that predictable but unobserved causes—the latent actions agents take—can explain the patterns of observed state factor transitions. While physical objects follow deterministic dynamics, agents' transitions result from internal decision-making processes that can be modeled as latent variables.

We incorporate this principle with a structured variational objective that aims to infer the actions that explain learned state transitions. For each learned state factor, VAD learns three interconnected components: a *policy* that captures an agent's action distribution, an *inverse dynamics model* that infers what action caused an observed transition, and a *forward dynamics model* that predicts the next state given an action. We show that this drives unsupervised learning of meaningful agent-oriented state factors purely from visual observations.

Our approach builds on structured representation learning, specifically slot-based attention mechanisms that decompose scenes into slot representations without supervision (Locatello et al., 2020; Kipf et al., 2021; Wu et al., 2023b; Wang et al., 2025), and extends beyond existing causal representation methods (Lippe et al., 2023) that identify binary interventions to model continuous agent policies. VAD provides the critical step, discovering agent representations that enable downstream tasks essential for ILFO, including action prediction and goal inference. Moreover, our learned representations exhibit properties consistent with biological agency perception and Theory of Mind (Rabinowitz et al., 2018; Oguntola et al., 2023), including shared action representations across agents (Cracco et al., 2019)—neural patterns that activate for the same action regardless of which agent performs it.

Our **contributions** are threefold:

1. A novel deep learning algorithm, VAD, which leverages a novel structured variational objective to imbue a structured neural network with the ability to discover agent-oriented state factors directly from visual observations.

2. Empirical demonstration that VAD discovers generalizable agent representations, maintaining high performance on novel agents, novel goals, and spatially-transformed coordination tasks while baselines drop to near-chance levels.

3. Evidence that learned representations exhibit properties consistent with human agency perception, including agent-invariant action encoding and teleological reasoning capabilities.

## 2 RELATED WORK

**Imitation Learning from Observation.** Recent ILFO methods learn policies from state-only demonstrations but assume pre-identified agents. Adversarial approaches like GAIfO (Torabi et al., 2018b) match state transition distributions, while variational methods (Edwards et al., 2019; Huang et al., 2024) infer discrete or continuous latent actions from observed transitions. Recent work extends ILFO to handle distribution shifts: goal-proximity methods (Lee et al., 2021) use learned progress estimators for generalization, domain-adaptive approaches (Choi et al., 2023) extract domain-invariant behavioral features across different embodiments, and offline pretrained models (Zhou et al., 2023) learn state abstractions from unlabeled video. All these methods require knowing which entities are agents a priori. VAD discovers agents without supervision, providing the missing prerequisite for ILFO in multi-agent scenes. Our variational framework shares conceptual similarity with recent probabilistic ILFO (Huang et al., 2024) but operates on multiple entities simultaneously without requiring action grounding.

**Object-Centric Learning and Agent Modeling.** Slot-based methods decompose scenes into entity representations: Slot Attention (Locatello et al., 2020) and SAVi (Kipf et al., 2021) learn object-centric representations, while SlotFormer (Wu et al., 2023b), SOLD (Mosbach et al., 2024), and Dyn-O (Wang et al., 2025) extend these to dynamics modeling with improved long-term prediction. These methods predict slots then decode to pixels, differing primarily in how they model slot dynamics $p(s_{t+1}|s_t)$. However, these approaches treat all entities uniformly. Recent work introduces structure to distinguish entity types: BISCUIT (Lippe et al., 2023) identifies latent causal variables and binary intervention indicators, discovering when external agents affect objects. In multi-agent settings, Theory of Mind approaches (Oguntola et al., 2023; Rabinowitz et al., 2018) model other agents' hidden states and beliefs, while opponent modeling methods predict agent policies for improved coordination. VAD combines these areas by augmenting slots with a variational objective that encourages agent-centric specialization through latent action inference. Unlike BISCUIT's binary interventions, we model continuous policies; unlike ToM approaches requiring known agents, we discover them directly from pixels.

## 3 METHOD

We formulate agent discovery as variational inference over latent actions that explain observed state transitions in an attention-based structured neural network. Our key insight: agents differ from objects through internal decision-making processes that drive their behavior. By modeling these latent actions explicitly, we create an inductive bias toward learning representations that capture agency.

### 3.1 PROBLEM FORMULATION

Given only visual observations $\mathbf{X} = \{x_0, x_1, ..., x_T\}$, where each frame $x_t \in \mathbb{R}^{H \times W \times C}$ contains multiple entities, we aim to learn slot-based representations that encode agent-specific information. Each frame is decomposed into $K$ slot representations $s_t^i$ (where $i \in \{1, ..., K\}$), with each slot potentially corresponding to an entity in the scene. We denote the set of all slot representations at time $t$ as $\mathcal{S}_t = \{s_t^1, ..., s_t^K\}$ and the set of all latent actions as $\mathcal{A}_t = \{a_t^1, ..., a_t^K\}$.

We make four key assumptions: (1) agents follow coherent policies modeled as conditional distributions $p(a^i|s^i)$ over actions $a^i$; (2) The Slot Attention architecture provides reasonable learned state factors to bootstrap learning of agent-oriented state factors; (3) observed transitions result from unobservable actions that must be inferred; and (4) these dynamics arise from agents executing actions according to their policies.

### 3.2 VARIATIONAL AGENT DISCOVERY

We model each slot's transition as potentially driven by a latent action, where agents take purposeful actions while non-agents follow environmental dynamics. To learn these latent actions and identify which slots represent agents, we formulate the problem as variational inference. The true joint transition probability requires intractable marginalization over all latent actions:

$$p(\mathcal{S}_{t+1}|\mathcal{S}_t) = \int p(\mathcal{S}_{t+1}|\mathcal{S}_t, \mathcal{A}_t)p(\mathcal{A}_t|\mathcal{S}_t)d\mathcal{A}_t. \tag{1}$$

To make this tractable, we apply variational inference, introducing a variational posterior $q_\theta(\mathcal{A}_t|\mathcal{S}_t, \mathcal{S}_{t+1})$ to approximate the true posterior and deriving an Evidence Lower BOund (ELBO) that provides a lower bound on the log marginal likelihood. To derive a tractable ELBO, we introduce factorized models. The forward dynamics factorizes as each slot depending on all current slots but only its own action:

$$p_\theta(\mathcal{S}_{t+1}|\mathcal{S}_t, \mathcal{A}_t) = \prod_{i=1}^{K} p_\theta(s_{t+1}^i|\mathcal{S}_t, a_t^i). \tag{2}$$

The inverse dynamics and policy prior also factorize:

$$q_\theta(\mathcal{A}_t|\mathcal{S}_t, \mathcal{S}_{t+1}) = \prod_{i=1}^{K} q_\theta(a_t^i|\mathcal{S}_t, s_{t+1}^i), \quad \pi_\theta(\mathcal{A}_t|\mathcal{S}_t) = \prod_{i=1}^{K} \pi_\theta(a_t^i|\mathcal{S}_t). \tag{3}$$

This yields the ELBO:

$$\mathcal{L}_{\text{ELBO}} = \sum_{i=1}^{K} \mathbb{E}_{q_\theta(a_t^i|\mathcal{S}_t, s_{t+1}^i)} \left[ \log p_\theta(s_{t+1}^i|\mathcal{S}_t, a_t^i) \right] - D_{KL}(q_\theta(a_t^i|\mathcal{S}_t, s_{t+1}^i)||\pi_\theta(a_t^i|\mathcal{S}_t)). \tag{4}$$

This decomposes into three components: A forward dynamics model $p_\theta(s_{t+1}^i|\mathcal{S}_t, a_t^i)$ that predicts the next slot state given all current slots and its action; an inverse dynamics model $q_\theta(a_t^i|\mathcal{S}_t, s_{t+1}^i)$ that infers actions from observed transitions; a learned policy prior $\pi_\theta(a_t^i|\mathcal{S}_t)$ representing the agent's action distribution given the global state.

The first term $\mathbb{E}_{q_\theta}[\log p_\theta(s_{t+1}^i|\mathcal{S}_t, a_t^i)]$ is the expected negative log-likelihood of the forward dynamics (which we denote $\mathcal{L}_{\text{forward}}$), measuring how well we predict next states given inferred actions. The KL divergence term $D_{KL}(q_\theta(a_t^i|\mathcal{S}_t, s_{t+1}^i)||\pi_\theta(a_t^i|\mathcal{S}_t))$ regularizes the inferred actions to be consistent with the learned policy.

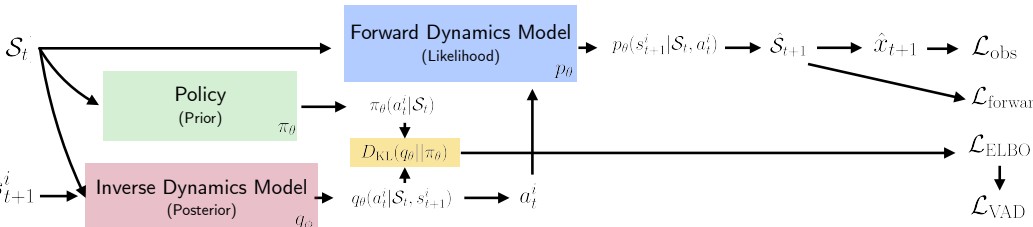

Figure 1: Given consecutive slot representations $s_t^i$ and $s_{t+1}^i$ from Slot Attention, three interconnected modules compute complementary objectives: (a) The Inverse Dynamics Model $q_\theta(a_t^i|\mathcal{S}_t, s_{t+1}^i)$ infers the latent action that caused the observed state transition, (b) The Forward Dynamics Model $p_\theta(s_{t+1}^i|\mathcal{S}_t, a_t^i)$ predicts the next slot state given all current states and the slot's action, contributing to the forward dynamics loss $\mathcal{L}_{\text{forward}} = -\log p_\theta(s_{t+1}^i|\mathcal{S}_t, a_t^i)$, and (c) The Policy Prior $\pi_\theta(a_t^i|\mathcal{S}_t)$ learns the agent's policy given the global state, regularizing the inferred actions through the KL divergence term.

Our full objective combines the ELBO across slots with observation reconstruction:

$$\mathcal{L}_{\text{VAD}} = \lambda_{\text{obs}}\mathcal{L}_{\text{obs}} + \lambda_{\text{ELBO}} \sum_{i \in \text{slots}} \mathcal{L}_i \tag{5}$$

where $\mathcal{L}_{\text{obs}} = -\log p_\theta(x_{t+1}|\hat{\mathcal{S}}_{t+1})$ is the observation reconstruction loss (negative log-likelihood), $\mathcal{L}_i$ is the per-slot ELBO from Equation 4, and $\lambda_{\text{obs}}, \lambda_{\text{ELBO}}$ are weighting hyperparameters. We summarize the algorithm for VAD in 1.

### 3.3 ARCHITECTURE

VAD consists of four abstract components that jointly learn to discover agents:

*State Encoder:* Encodes visual observations $x_t$ into structured state representations $\mathcal{S}_t = \{s_t^1, ..., s_t^K\}$ that decompose the scene into $K$ entity-centric components.

*Action Inference Module:* The inverse dynamics model $q_\theta(a_t^i|\mathcal{S}_t, s_{t+1}^i)$ infers latent actions from global state and slot transitions. The policy prior $\pi_\theta(a_t^i|\mathcal{S}_t)$ learns the agent's behavioral distribution over actions given the global state.

*Transition Model:* The forward dynamics model implements the factorized prediction $p_\theta(s_{t+1}^i|\mathcal{S}_t, a_t^i)$, where each slot's next state depends on the global state context $\mathcal{S}_t$ but only its own action $a_t^i$. This factorization naturally encourages specialization: slots representing agents learn coherent dynamics driven by their actions, while environment slots remain unaffected by action variables.

*State Decoder:* Reconstructs observations from predicted states $\hat{\mathcal{S}}_{t+1}$, computing the observation likelihood $p_\theta(x_{t+1}|\hat{\mathcal{S}}_{t+1})$.

Implementation details are provided in Appendix B.

## 4 AGENT REPRESENTATION LEARNING EXPERIMENTS

We evaluate whether VAD learns representations that capture agency by probing learned slots to predict agent actions and goals from their representations alone.

### 4.1 EXPERIMENTAL SETUP

*Evaluation Protocol.* For action prediction with VAD, we evaluate the learned policy prior $\pi_\theta(a_t^i|\mathcal{S}_t)$ by training a linear alignment layer that maps our learned action representations to ground-truth action

---

**Algorithm 1** Variational Agent Discovery (VAD)

---

**Require:** Observation sequence $\mathbf{X} = \{x_0, x_1, \ldots, x_T\}$, number of entities $K$
**Require:** Learning rate $\alpha$, loss weights $\lambda_{\text{obs}}, \lambda_{\text{ELBO}}$
**Ensure:** Trained parameters $\theta^*$
 1: Initialize state encoder, action inference modules ($q_\theta, \pi_\theta$), transition model $p_\theta$, state decoder
 2: **for** each training iteration **do**
 3:     Sample batch $\{\mathbf{X}^{(b)}\}_{b=1}^B$ from dataset
 4:     **for** each sequence $\mathbf{X}^{(b)}$ in batch **do**
 5:         *// Encode observations to states*
 6:         **for** $t = 0$ to $T - 1$ **do**
 7:             $\mathcal{S}_t \leftarrow \text{StateEncoder}(x_t, \mathcal{S}_{t-1})$ *// Obtain $K$ state factors*
 8:         **end for**
 9:         *// Compute variational objective*
10:         $\mathcal{L}_{\text{ELBO}}^{(b)} \leftarrow 0$
11:         **for** $t = 0$ to $T - 2$ **do**
12:             **for** each factor $i \in \{1, \ldots, K\}$ **do**
13:                 *// Infer latent action from global state and slot transition*
14:                 $a_t^i \sim q_\theta(a_t^i|\mathcal{S}_t, s_{t+1}^i)$ *// Posterior inference*
15:                 *// Predict next state with factorized dynamics*
16:                 $\hat{s}_{t+1}^i \leftarrow p_\theta(s_{t+1}^i|\mathcal{S}_t, a_t^i)$ *// Depends on all states but only own action*
17:                 *// Compute slot ELBO*
18:                 $\mathcal{L}_i \leftarrow \log p_\theta(s_{t+1}^i|\mathcal{S}_t, a_t^i) - D_{\text{KL}}(q_\theta(a_t^i|\mathcal{S}_t, s_{t+1}^i)||\pi_\theta(a_t^i|\mathcal{S}_t))$
19:                 $\mathcal{L}_{\text{ELBO}}^{(b)} \leftarrow \mathcal{L}_{\text{ELBO}}^{(b)} + \mathcal{L}_i$
20:             **end for**
21:             $\hat{\mathcal{S}}_{t+1} \leftarrow \{\hat{s}_{t+1}^1, ..., \hat{s}_{t+1}^K\}$ *// Collect predicted states*
22:             *// Decode and compute observation loss*
23:             $\mathcal{L}_{\text{obs}}^{(b)} \leftarrow -\log p_\theta(x_{t+1}|\hat{\mathcal{S}}_{t+1})$
24:         **end for**
25:     **end for**
26:     *// Update parameters*
27:     $\mathcal{L}_{\text{VAD}} \leftarrow \frac{1}{B}\sum_{b=1}^B (\lambda_{\text{obs}}\mathcal{L}_{\text{obs}}^{(b)} + \lambda_{\text{ELBO}}\mathcal{L}_{\text{ELBO}}^{(b)})$
28:     $\theta \leftarrow \theta - \alpha\nabla_\theta \mathcal{L}_{\text{VAD}}$
29: **end for**
30: **return** $\theta^*$

---

labels. This alignment is necessary because VAD learns action representations without supervision; the linear layer simply learns the correspondence between our discovered action categories and the environment's action space. For SAVi baseline, we train linear probes directly on slot representations. Since all $K$ slots have associated policies (including those corresponding to non-agents), we evaluate each slot's action predictions against each agent's true actions, rank slots by accuracy, and report the best match. For goal prediction, both methods train linear probes directly on slot representations, similarly ranking slots by accuracy and reporting the best match. This ranking procedure tests whether specific slots consistently specialize to track agents. We evaluate generalization using: novel goals (Minigrid: red key instead of green box), novel configurations (Overcooked: vertically flipped kitchen), and novel agents (MPE: third agent added at test time).

*Baselines.* We compare against SAVi (Kipf et al., 2021), a state-of-the-art object-centric model, which uses identical architecture (CNN + Slot Attention) but lacks our variational action modeling. For SAVi, we train linear probes directly on slot representations for action prediction (using the same ranking procedure), isolating the contribution of our latent action modeling.

### 4.2 MINIGRID: SINGLE-AGENT NAVIGATION

*Environment.* Minigrid (Chevalier-Boisvert et al., 2023) provides a discrete $10 \times 10$ grid world where a single agent (red triangle) navigates around obstacles to reach goal objects. The agent operates with four actions: forward, left, right, and pickup. During training, the agent pursues a green box goal; at test time, we evaluate on a novel red key goal.

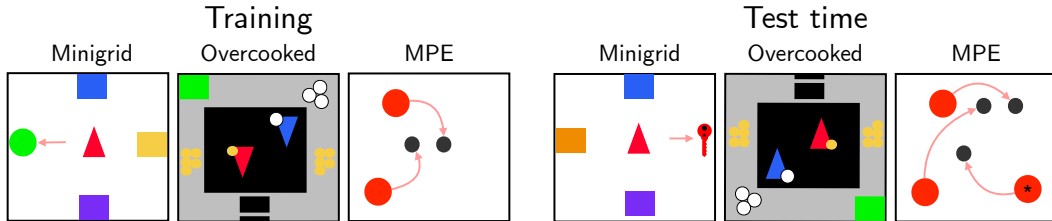

Figure 2: Evaluation environments. **Left:** Training configurations. **Right:** Test-time generalization scenarios. In Minigrid, the agent (red triangle) navigates to goals (green ball during training, red key at test) among distractor objects. In Overcooked, two agents (red/blue triangles) coordinate to cook onions (yellow) in pots (black), serve on plates (white), and deliver to the goal (green). In MPE, agents (red circles) must cover landmarks (black circles); a third agent appears only at test time.

Table 1: Minigrid results (%, mean $\pm$ SE over 12 runs). Best slot performance reported.

| Task | Model | Familiar Goal | Novel Goal |
|---|---|---|---|
| Action | SAVi | $62.1 \pm 0.9$ | $37.4 \pm 0.8$ |
| | **VAD** | $\mathbf{90.2 \pm 0.7}$ | $\mathbf{76.8 \pm 1.0}$ |
| Goal | SAVi | $55.3 \pm 0.7$ | $32.0 \pm 0.9$ |
| | **VAD** | $\mathbf{91.5 \pm 0.6}$ | $\mathbf{79.1 \pm 1.0}$ |

*Challenges.* The key generalization challenge is goal transfer—can the model maintain performance when the target category and location changes? This tests whether learned representations capture goal-directed behavior independent of specific visual features.

*Results.* Table 1 shows action and goal prediction accuracy. VAD achieves $90.2 \pm 2.3\%$ action prediction accuracy on familiar goals, maintaining $76.8 \pm 3.6\%$ on novel goals. In contrast, SAVi drops from $62.1 \pm 3.2\%$ to $37.4 \pm 2.8\%$. For goal inference, VAD maintains $79.1 \pm 3.4\%$ accuracy on novel goals versus SAVi's $32.0 \pm 3.1\%$. VAD's substantial absolute advantage demonstrates that modeling latent actions creates representations robust to goal changes.

### 4.3 OVERCOOKED: TWO-AGENT COORDINATION

*Environment.* Overcooked (Carroll et al., 2020) simulates a cooperative cooking game where two agents (red and blue triangles) coordinate in a virtual kitchen. Agents must collect ingredients from counters, cook them in pots, place on plates, and deliver to serving areas. The action space includes five actions: forward, left, right, pickup, and putdown.

*Challenges.* The main challenge is spatial generalization—at test time, the kitchen layout is vertically flipped, requiring agents to adapt their coordination strategies. This tests whether representations encode abstract roles (e.g., "ingredient collector" vs "cook") rather than location-specific behaviors.

*Results.* Table 2 shows per-agent performance. For action prediction, VAD achieves approximately 80% accuracy for both agents in familiar configurations, maintaining 70-72% in novel layouts. SAVi shows larger degradation: from 57-59% to 39-41%. For goal prediction (identifying which dishes to deliver), VAD maintains 87% accuracy in familiar layouts and 74% in novel configurations, while SAVi achieves 75% familiar but drops to 48% in novel layouts. The consistent performance across both agents and tasks suggests VAD learns role-based representations that transfer across spatial transformations.

### 4.4 MULTI-AGENT PARTICLE ENVIRONMENT: NOVEL AGENT DISCOVERY

*Environment.* MPE (Lowe et al., 2020) implements a cooperative coverage task where circular agents must reach landmark goals while avoiding collisions. Agents move with five actions (up, down, left, right, stay) and receive global rewards for landmark coverage minus collision penalties.

Table 2: Overcooked results (%, mean $\pm$ SE over 12 runs). Per-agent performance.

| Task | Model | Agent 1 | | Agent 2 | |
|---|---|---|---|---|---|
| | | Familiar | Novel | Familiar | Novel |
| Action | SAVi | $58.7 \pm 1.0$ | $41.2 \pm 0.8$ | $56.4 \pm 1.1$ | $39.1 \pm 0.9$ |
| | **VAD** | $\mathbf{81.3 \pm 0.7}$ | $\mathbf{71.9 \pm 1.1}$ | $\mathbf{79.8 \pm 0.8}$ | $\mathbf{70.2 \pm 1.0}$ |
| Goal | SAVi | $75.2 \pm 0.5$ | $48.3 \pm 1.2$ | $74.8 \pm 0.8$ | $47.1 \pm 0.9$ |
| | **VAD** | $\mathbf{87.4 \pm 0.7}$ | $\mathbf{74.6 \pm 1.0}$ | $\mathbf{86.9 \pm 0.5}$ | $\mathbf{73.8 \pm 0.8}$ |

Table 3: MPE results (%, mean $\pm$ SE over 12 runs). Per-agent performance with novel agent.

| Task | Model | Agent 1 | Agent 2 | Agent 3* |
|---|---|---|---|---|
| Action | SAVi | $61.2 \pm 1.0$ | $59.0 \pm 0.8$ | $42.3 \pm 1.1$ |
| | **VAD** | $\mathbf{81.7 \pm 0.8}$ | $\mathbf{80.3 \pm 0.7}$ | $\mathbf{71.4 \pm 1.1}$ |
| Goal | SAVi | $53.6 \pm 1.0$ | $52.8 \pm 0.9$ | $38.4 \pm 1.2$ |
| | **VAD** | $\mathbf{86.9 \pm 0.6}$ | $\mathbf{87.3 \pm 0.7}$ | $\mathbf{84.1 \pm 0.9}$ |

*Novel agent not seen during training.

*Challenges.* The critical test is novel agent generalization—a third agent appears at test time that was never observed during training. This evaluates whether the model can discover and track new agents zero-shot, assigning them to unused slots while preserving existing agent representations.

*Results.* Table 3 shows per-agent performance. For familiar agents (1-2), VAD achieves 80-82% action prediction and 87% goal prediction accuracy. Crucially, for the novel third agent, VAD maintains 71% action and 84% goal accuracy. SAVi shows catastrophic degradation for the novel agent: dropping to 42% for actions and 38% for goals. Figure 3 visualizes how VAD assigns the new agent to slot 3 while preserving slots 1-2 for familiar agents, demonstrating compositional generalization.

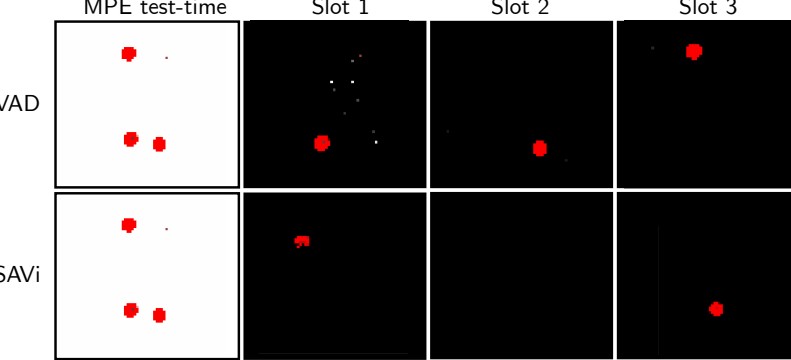

Figure 3: Slot specialization in MPE at test time with three agents. Top row: VAD successfully reconstructs all three agents in slots 1, 2, and 3. Bottom row: SAVi struggles with the novel third agent, only representing two agents despite three being present.

## 5 RECAPITULATING FINDINGS FROM NEUROSCIENCE AND PSYCHOLOGY

Beyond studying whether VAD enables unsupervised learning of agent-oriented representations in machines, we see if it can also recapitulate findings from neuroscience and developmental psychology.

## 5.1 SHARED ACTION REPRESENTATIONS ACROSS MULTIPLE AGENTS

Here, we aim to recapitulate findings on multi-agent action observation (Cracco et al., 2019). In the MPE environment with three agents, we analyze the 128-dimensional slot vectors $s_t^i$ conditioned on agent actions. For each action $a \in \{\text{Left}, \text{Right}, \text{Up}, \text{Down}, \text{Stay}\}$, we group slot activations: $\mathcal{S}_{a,i} = \{s_t^i \mid a_t^i = a\}$. We collect all the slot representations for agent $i$ whenever it performs action $a$, then check if different agents exhibit similar values in their 128-dimensional feature vectors when performing the same action. We compute cross-agent correlations for each feature dimension $j$:

$$r_{a,j}^{i,k} = \text{Pearson}(\mathcal{S}_{a,i}^j, \mathcal{S}_{a,k}^j) \tag{6}$$

where $\mathcal{S}_{a,i}^j$ denotes feature $j$ of agent $i$'s slot when performing action $a$.

Figure 4 demonstrates agent-invariant action encoding for the Right action. Feature 57 exhibits consistent activation (correlation $r > 0.85$) across all agent pairs when they perform the same action. The 3D scatter plot (middle panel) shows tight clustering of feature values across agents, indicating shared activation patterns. Similar analysis reveals feature 107 for Up actions and feature 78 for Down actions show analogous shared representations (see Appendix A for complete analysis).

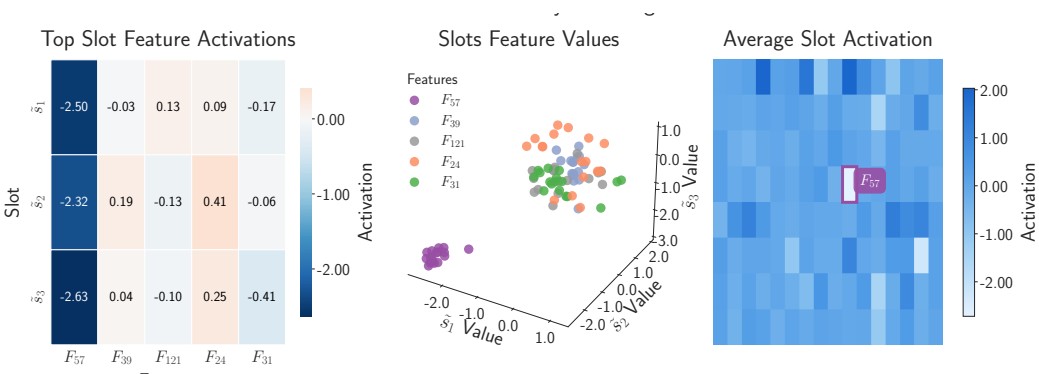

Figure 4: Shared action representation analysis for Right action. Left: Feature activation heatmap for top five features showing cross-agent correlations. Middle: 3D scatter plot of feature values across agents (same color = same feature). Right: Average slot activations with feature F57 (strongest cross-agent correlation) highlighted.

These shared action representations emerge without supervision, suggesting VAD develops a common action encoding that abstracts away from agent-specific details. This parallels findings from human neuroscience showing that observing multiple agents' actions activates overlapping neural representations in the motor system (Cracco et al., 2019). The action-specificity of these patterns—distinct features for each action—indicates specialized encodings rather than general movement detection, enabling VAD's strong generalization to novel agents.

## 5.2 TELEOLOGICAL REASONING AND RATIONAL ACTION

Inspired by Gergely & Csibra (2003)'s infant cognition studies, we tested whether VAD captures principles of rational agency. After training on an agent jumping over an obstacle to reach a goal, we removed the obstacle and asked the model to predict future trajectories.

As shown in Figure 5, VAD correctly predicted direct paths when obstacles disappeared—behavior that 12-month-old infants also expect. This suggests VAD learns more than statistical correlations; it appears to disentangle goals from the means to achieve them, enabling rational predictions in novel scenarios.

## 6 DISCUSSION

VAD demonstrates that variational inference over latent actions provides a powerful inductive bias for discovering agent-centric representations from pixels. Our experiments (Section 4) show VAD

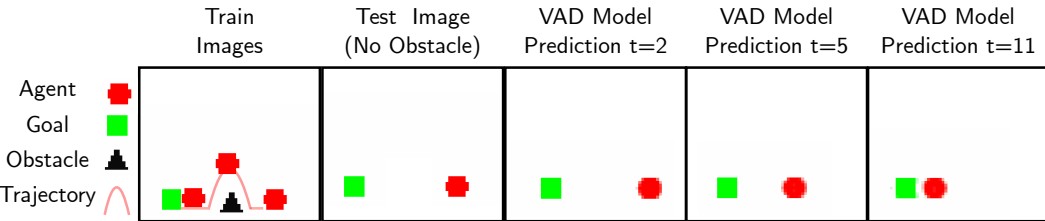

Figure 5: Rational action prediction. (1) Training: agent jumps over obstacle, (2) Test: obstacle removed, (3-5) VAD predictions at $t = 2, 5, 11$. Despite never seeing obstacle-free paths during training, VAD predicts direct trajectories, demonstrating teleological reasoning.

achieves strong performance on downstream tasks: action prediction accuracy of 90.2% (Minigrid), 80.5% (Overcooked averaged), and 81% (MPE averaged), alongside goal inference accuracy of 91.5%, 87.2%, and 87.1% respectively. Crucially, these representations generalize to novel agents (71.4% action accuracy on unseen third agent), novel goals (76.8% on new target objects), and novel configurations (71% on flipped layouts).

Beyond task performance, VAD exhibits emergent cognitive properties. The shared action representations discovered across agents parallel recent neuroscience findings showing humans simultaneously represent multiple observed actions through overlapping neural patterns (Cracco et al., 2019). Additionally, VAD demonstrates teleological reasoning capabilities similar to 12-month-old infants, predicting rational, direct paths when obstacles are removed despite never observing such trajectories during training. Notably, current vision-language models struggle with similar visual theory of mind tasks: Schulze Buschoff et al. (2025) found that in intuitive psychology tasks involving 2D geometric agents—similar to our evaluation environments—none of the tested VLMs showed strong alignment with human judgments, highlighting the challenge of learning agency representations even for state-of-the-art multimodal models.

**Limitations.** While VAD shows strong empirical results, important limitations remain. Most critically, VAD does not provide explicit thresholds for distinguishing agents from non-agents, agency is inherently subjective and context-dependent. Our method discovers agent-oriented representations without categorical boundaries, though these representations significantly improve downstream agent-based tasks. Additionally, our linear probing evaluation may not capture all learned structure, and our environments use optimal policies whereas real-world agents exhibit stochastic, suboptimal behavior. The current evaluation focuses on controlled environments with clear agent-object distinctions; naturalistic settings with ambiguous agency present additional challenges. Future work could explore continuous action spaces via normalizing flows, hierarchical action representations for complex behaviors, and applications to real video where ground-truth actions are unavailable.

## 7 CONCLUSION

We introduced Variational Agent Discovery (VAD), a method that learns agent-centric representations from pixels through variational inference over latent actions. By modeling state transitions as consequences of unobservable agent decisions, VAD discovers representations that capture agency structure rather than visual patterns.

Our experiments demonstrate that VAD substantially outperforms object-centric baselines across diverse generalization scenarios—novel agents, goals, and environments—while exhibiting emergent properties like shared action representations and teleological reasoning. These results suggest that modeling latent actions provides a useful inductive bias for learning agent-oriented representations from pixels, enabling better generalization without requiring labeled data.

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

## A  SHARED ACTION REPRESENTATION ANALYSIS DETAILS

### A.1  METHODOLOGY

To investigate shared action representations in VAD's learned features, we perform detailed analysis of slot vector activations conditioned on agent actions in the MPE environment. Each slot representation $s_t^i$ is a 128-dimensional vector. We manually verified slot-to-agent mappings across 205 episodes, confirming consistent assignments.

For each action $a$ and feature dimension $j$, we compute shared action scores by averaging absolute correlation coefficients across all agent pairs:

$$\text{SharedActionScore}_{a,j} = \frac{1}{|P|} \sum_{(i,k) \in P} |r_{a,j}^{i,k}| \tag{7}$$

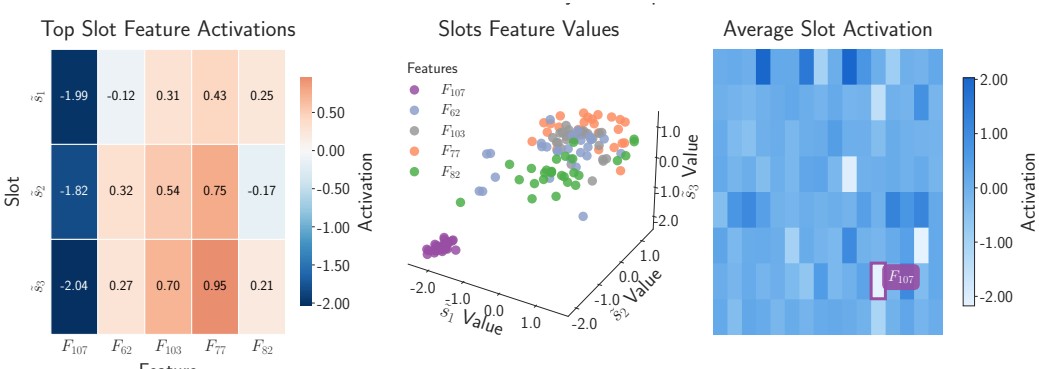

Figure 6: Shared action representation analysis for Up action. Feature F107 shows the strongest cross-agent correlations with highly consistent activation patterns across all three agent slots during upward movement.

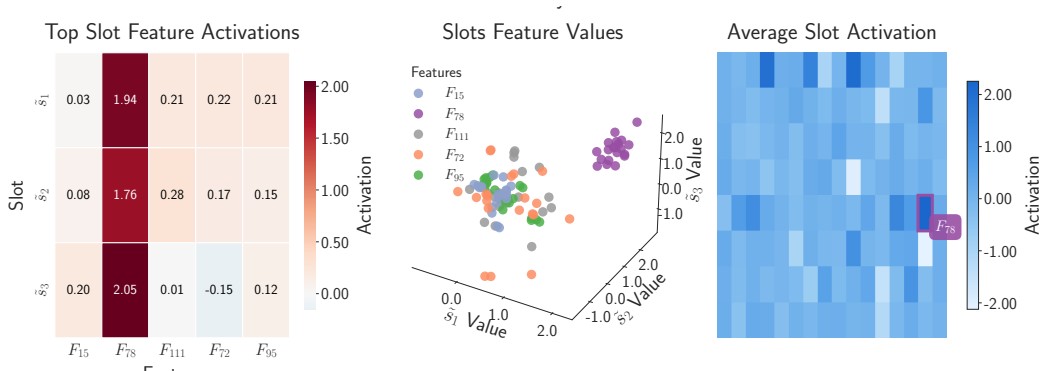

Figure 7: Shared action representation analysis for Down action. Feature F78 exhibits the strongest cross-agent correlations, with consistently positive activation values across all agent slots during downward movement.

where $P$ is the set of all agent pairs and $r_{a,j}^{i,k}$ is the Pearson correlation between feature $j$ of agents $i$ and $k$ when performing action $a$.

### A.2 COMPLETE RESULTS

Our analysis reveals distinct shared representations for each action type, with specific features showing strong cross-agent correlations:

### A.3 INTERPRETATION

The emergence of action-specific shared features without explicit supervision suggests fundamental computational principles:

**Action Abstraction.** The inverse dynamics model $q_\theta(a|s_t, s_{t+1})$ must infer actions from state transitions regardless of which agent performs them. This creates pressure for learning action representations that abstract away from agent-specific details while preserving action-relevant information.

**Specialized Encoding.** Each action develops distinct feature signatures: F57 for Right (Figure 4 in main text), F107 for Up (Figure 6), F78 for Down (Figure 7). The Left action shows distributed encoding (Figure 8), while Stay combines multiple features (Figure 9). This specialization suggests the model learns meaningful action categories rather than generic movement detection.

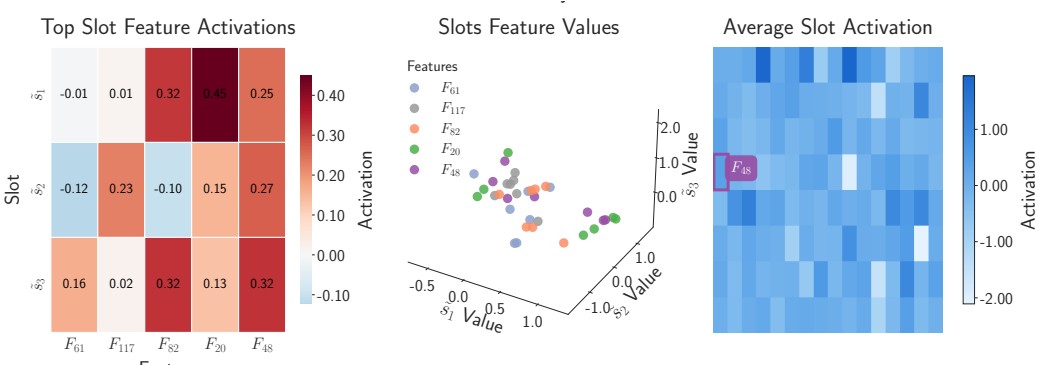

Figure 8: Shared action representation analysis for Left action. Unlike other directional actions, leftward movement is represented by multiple features with moderate correlations rather than a single strongly correlated feature.

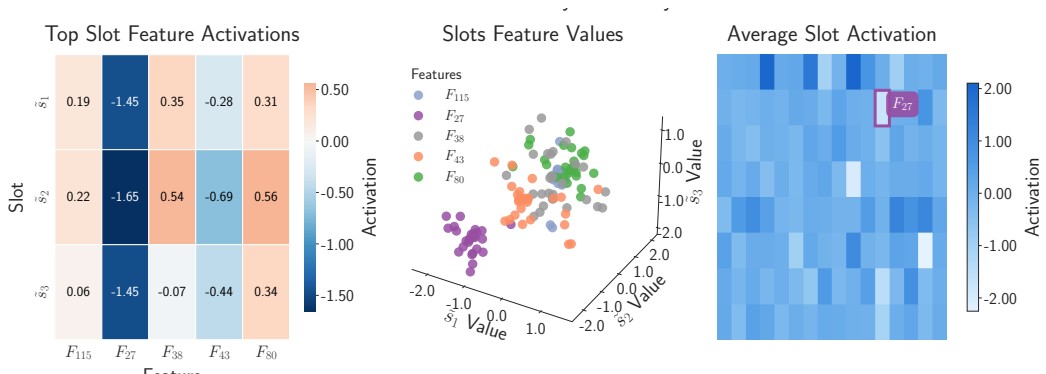

Figure 9: Shared action representation analysis for Stay action. The stationary state is encoded through a combination of features, including negative activation of F27 and F43 alongside positive activation of F80.

**Cross-Agent Generalization.** The high correlations ($r > 0.85$ for Right, $r > 0.80$ for Up/Down) across agent pairs indicate robust action encoding that generalizes across entities. This property is crucial for the strong generalization to novel agents demonstrated in our quantitative results (Tables 1-2 in main text).

**Computational Parallels.** The spontaneous emergence of these patterns from our variational objective parallels recent findings in neuroscience showing that humans can simultaneously represent multiple observed actions through overlapping neural patterns (Cracco et al., 2019). Both artificial and biological systems appear to solve the fundamental problem of multi-agent action observation through shared representational codes.

# B  IMPLEMENTATION DETAILS

## B.1  ARCHITECTURE DETAILS

Our implementation instantiates the abstract VAD architecture as follows:

**State Encoder:** We use SAVi (Kipf et al., 2021) with a CNN backbone processing $64 \times 64$ RGB images. Slot Attention with $K = 5$ slots (128-dimensional) decomposes scenes using 3 iterations

during training. We employ implicit differentiation (Wu et al., 2023a), detaching gradients from all but the final iteration for stable training.

**Action Inference:** The inverse dynamics model $q_\theta$ is a 2-layer MLP with GELU activations, processing concatenated slot pairs $(s_t^i, s_{t+1}^i)$ to output action logits. For discrete action spaces, we use Gumbel-Softmax (Jang et al., 2016; Maddison et al., 2016) with temperature annealing from 1.0 to 0.1. The policy prior $\pi_\theta$ is similarly structured.

**Transition Model (Factorized):** The forward dynamics implements $p_\theta(s_{t+1}^i | \mathcal{S}_t, a_t^i)$ using a Transformer architecture. For each slot $i$, we construct tokens by concatenating each slot with either the inferred action (for slot $i$) or zeros (for other slots): $[s_t^j | a_t^j]$ where $a_t^j = a_t^i$ if $j = i$, else $\mathbf{0}$. This enforces the factorization where each slot's dynamics depend on all current states but only its own action. A relational Transformer then processes these tokens to predict next states.

**State Decoder:** A spatial broadcast decoder (Watters et al., 2019) projects slots to spatial feature maps, adds position embeddings (Fourier features), processes through a CNN, and generates per-slot reconstructions with alpha masks for compositing.

## B.2 TRAINING DETAILS

**Optimization:** Adam optimizer, learning rate $4 \times 10^{-4}$, batch size varies by environment, 100K gradient steps.

**Loss Components:**

- Reconstruction loss: L2 distance between predicted and actual next frame
- KL divergence: Between posterior $q_\theta(a|s_t, s_{t+1})$ and learned prior $\pi_\theta(a|s_t)$
- Loss weights: $\lambda_{\text{obs}} = 1.0$, $\lambda_{\text{ELBO}}$ annealed during training

**Gumbel-Softmax:** Temperature parameter for differentiable discrete sampling, annealed from 1.0 to lower values during training to approach hard categorical sampling.

**Data:** Episodes collected from optimal or near-optimal policies in each environment. Sequences processed as consecutive frame pairs for learning transitions. Train/val/test splits vary by environment with held-out scenarios for generalization testing.

## B.3 EVALUATION PROTOCOL

**Action Prediction:** For VAD, we evaluate the learned policy prior $\pi_\theta(a|s_t)$ by training a linear alignment layer that maps our unsupervised action representations to ground-truth action labels. This is a simple affine transformation that learns the correspondence between VAD's discovered action categories (5-dimensional distributions) and the environment's true action space. For the SAVi baseline, we train linear probes directly on slot representations $s_t^i$ (128-dimensional vectors) since it lacks explicit action modeling. Both methods use identical ranking: evaluate each slot's predictions against each agent's ground-truth actions, report best match per agent.

**Goal Prediction:** Both VAD and SAVi train linear probes $\hat{g}_t^i = \text{softmax}(W_g s_t^i + b_g)$ on slot representations to classify which goal each slot is pursuing. Same ranking procedure as action prediction.

## B.4 MODEL COMPONENTS

**Conditional VAE (CVAE):** Core variational component with:

- Inverse dynamics: MLP with 2 hidden layers, GELU activations, variance-scaled initialization
- Forward dynamics: Transformer processing concatenated action-slot representations
- Learned prior: Trainable categorical distribution parameters initialized with small random values
- Gumbel-Softmax: Differentiable sampling for discrete actions during training

**SAVI Module:** Processes CNN features through iterative slot refinement:

- Inverted attention: Softmax over query (slot) axis rather than key axis
- GRU cell: Updates slots based on attention-weighted features
- Layer normalization: Applied to queries before attention computation
- Implicit differentiation: Gradients flow only through final iteration

**Generalization Scenarios:**

- Minigrid: Novel goal (red key vs green box during training)
- Overcooked: Novel configuration (vertically flipped kitchen layout)
- MPE: Novel agent (third agent added at test time, not seen during training)

## C   LLM USAGE DISCLOSURE

In accordance with ICLR 2026 policy, we disclose that large language models were used for minor text polishing, grammar correction, and formatting assistance during manuscript preparation. No LLMs were used for generating scientific ideas, experimental design, or result interpretation.

