# OpenReview forum: "Variational Agent Discovery"
_ICLR.cc/2026/Conference — ICLR 2026 Conference Withdrawn Submission_

### Official Review · Reviewer_YVaZ · 2025-10-28

**Soundness:** 2
**Presentation:** 3
**Contribution:** 2
**Rating:** 4
**Confidence:** 4

**Summary:**

This paper introduces VAD, an unsupervised method for discovering agent-centric representations from pixel observations. VAD augments slot attention with a structured variational objective that jointly learns inverse dynamics, forward dynamics, and policy priors to distinguish agents from objects. The authors demonstrate improved performance over SAVi on action prediction and goal inference tasks across three environments, and show emergent properties like shared action representations and teleological reasoning.

**Strengths:**

- It's a very well motivated problem. Distinguishing agents from passive objects in visual scenes without supervision is an important and under-explored challenge, particularly for downstream tasks like imitation learning from observation.
- The method shows good and consistent performance across three diverse environments with different numbers of agents and action spaces. VAD maintains good accuracy even on novel scenarios where the SAVi drops significantly. The novel agent experiment is particularly nice, VAD can discover and track a third agent never seen during training while SAVi is not able to do much on this task.
- The method also shows some interesting emergent properties, for instance, shared action representations where specific features activate consistently across different agents, teleological reasoning predicting rational paths when obstacles are removed, etc.

**Weaknesses:**

- The architecture is essentially SAVi plus standard variational inference components (inverse dynamics, forward dynamics, policy prior). While the factorized structure is interesting, individual components are well-established in prior work. The main novelty is applying factorized variational inference to multiple slots simultaneously, making the work rather incremental. However, given the interesting conclusions and experiments, this is a minor concern.
- **This is in my understanding is a critical weakness.** The paper only compares to SAVi, which has no action modeling at all. This leaves a large gap between "no action awareness" (SAVi) and "full variational framework," making it hard for me to assess what's actually driving the strong performance. Below I list some interesting baselines that could have made the paper stronger in my opinion:
    - Add a simple classifier on top of SAVi slots that directly predicts actions using ground-truth labels during training. If this simple supervised approach matches VAD's performance, then the complex unsupervised variational framework might not be needed.
   - Change VAD so each slot's next state depends on ALL actions:  p(s^i_{t+1} | S_t, A_t) instead of p(s^i_{t+1} | S_t, a^i_t) The paper claims factorization drives agent-specific slot specialization, i.e., each slot  learns to represent an agent because it only sees that agent's actions. If non-factorized VAD works just as well, the narrative of the work will weaken.
   - While the paper distinguishes itself from SlotFormer (ICLR 2023) by noting that SlotFormer lacks action modeling, SlotFormer already extends SAVi with Transformer-based dynamics. I am not very sure about this point but perhaps SlotFormer can be easily augmented with action prediction heads?  such a model will be a direct comparison to VAD.
- Ablations: The paper lacks component ablations to determine which parts of the system are actually necessary. The method combines three components, inverse dynamics, forward dynamics, and policy prior, but provides no evidence that all three are needed. Testing VAD without the KL term (keeping only inverse and forward dynamics) would show whether the policy prior is actually necessary or redundant. Testing without forward dynamics (keeping only inverse dynamics and policy) would reveal whether state prediction is crucial or if learning action distributions alone suffices. Testing without inverse dynamics (using ground-truth actions during training) would determine whether unsupervised action inference is essential. Without these ablations, it is rather difficult to assess what is leading to the success of the method - maybe two components would work just as well, or maybe one is doing most of the heavy lifting.

**Questions:**

1. Can you provide results for a version where each slot's next state depends on ALL actions (p(s^i_{t+1} | S_t, A_t)) instead of only its own action? This can be a test for one of your main claims that factorization drives agent-specific slot specialization.

2. How does SAVi perform when augmented with a simple supervised action prediction head (linear classifier with ground-truth labels)?

3. Which components of your three-part system (inverse dynamics, forward dynamics, policy prior) are actually essential? Can you provide ablations showing performance when removing each component individually?

4. SlotFormer already extends SAVi with Transformer-based dynamics. Could you compare to SlotFormer augmented with action prediction, or explain why this wouldn't be a suitable baseline for your task?

---

### Official Review · Reviewer_iuzA · 2025-10-30

**Soundness:** 2
**Presentation:** 3
**Contribution:** 2
**Rating:** 4
**Confidence:** 3

**Summary:**

This paper introduces an unsupervised learning method, Variational Agent Discovery (VAD), for learning agent-oriented action representations, and hence discovering agents, from visual observations. VAD uses slot-based attention with three interconnected components: (1) an inverse dynamics model that infers actions from state transitions, (2) a forward dynamics model that predicts next states from actions, and (3) a policy prior representing agent action distributions. The method is evaluated on three environments (Minigrid, Overcooked, MPE) showing strong generalization to novel agents, goals, and configurations. The authors also demonstrate shared action representations across agents and "teleological reasoning capabilities".

**Strengths:**

- Framing agent discovery as a variational inference problem results in a compelling mathematical framework and the core insight of distinguishing agents from objects through internal decision-making processes that can be modelled as latent variables is promising. The ELBO derivation is well-presented.
- The paper demonstrates strong generalisation results across novel agents (Table 3), novel goals (Table 1), and novel configurations (Table 2), when compared against the SAVi baseline.
- The studies on the connections to developmental psychology and neuroscience are interesting: the shared action representation analysis (Section 5.1) reveals cross-agent correlations $r > 0.85$ for the same action, and VAD is shown to predict direct object paths in the presence of obstacles despite not being trained on obstacle-free paths (Figure 5).

**Weaknesses:**

- The ELBO doesn't seem to explicitly encourage agent vs. non-agent differentiation. All $K$ slots have associated policies, including those corresponding to static objects. It's not clear to me what prevents non-agent slots from learning spurious action distributions, how can this be tested? This is especially important since I do not see any identifiability analysis for the latent action variable. Multiple agent representations can yield the same ELBO.
- How can we be sure that slot attention provides reasonable object-like factors at the start? How is this coherent with unsupervised agent discovery?
- I might have missed this, but: what is the justification behind the factorisation assumed in Equation 2, or can you comment on what cases it cannot address? What would happen in the case of multiple agents, as present in Overcooked and MPE environments, where intuitively the action of one agent should depend on the other's action? How are contacts dealt with?
- VAD does not provide explicit thresholds for agent identification-- then how do we know if a slot indeed represents an agent, or is merely learns some useful dynamics? Moreover, it seems the evaluation relies on manually verifying slot-to-agent mappings and the ranking procedure assumes we know which slots are agents. This makes the claim of _discovering_ agents seem overstated. What are your thoughts on this?
- The scalability of the proposed method is questionable since all considered environments are simple 2D with clear agent-object distinctions and optimal agent policies. Even if 3D environments are out of scope, what about 2D environments with partial observability or noisy observations? Would the method scale to stochastic agents or agents with suboptimal policies? The connection to "real tutorial videos or gameplay demonstrations" in the introduction seems like a stretch in the absence of clarity on these questions.
- How does the method compare against a supervised baseline with agent labels? Also, why are comparisons to the cited methods: BISCUIT, SlotFormer, and Dyn-O not considered?
- The evaluations would benefit from more details: what is the schedule for temperature annealing? what is the sensitivity to $\lambda_{obs}$ and $\lambda_{ELBO}$? does SAVi also show similar behaviour as the one observed from VAD in Figure 5?



I am happy to reconsider my score if my issues and questions can be addressed.

**Questions:**

- What empirically prevents static object slots from learning meaningful action distributions? Is there analysis of what actions these slots _learn_?
- How does generalisation to a novel agent work? Does VAD use a previously unused slot, or reassign existing slots dynamically?
- How is the claim for the emergence of cognitive phenomena justified?
- When does VAD fail-- can VAD handle agents controlling multiple objects, or partially observable agents, or adversarial agents, or agents with suboptimal policies?

---

### Official Review · Reviewer_HoVg · 2025-10-31

**Soundness:** 4
**Presentation:** 3
**Contribution:** 2
**Rating:** 4
**Confidence:** 4

**Summary:**

The paper proposes an action conditioned slot attention method for discovering agentic representation, i.e. latent actions per slots/object that can explain the observed dynamics. The training is unsupervised.
The evaluation is on the task of action prediction in three toy environments. Also further analysis and relation to findings in neuroscience are made.

**Strengths:**

- Clean approach to incorporate latent actions
- Logical presentation
- Good results

**Weaknesses:**

- limited evaluation, 3 toy environments, but they are all static
- no ILFO done: the paper mentions the importance of the work for imitation learning from observations, but is not showing any results on that.
- more analysis would be great, see questions
- shared action representation results are anecdotal.

**Questions:**

Q1: Is the Slot attention model trained end-to-end? In the algorithm box, it is not clear whether the StateEncoder is also trained?

Q2: Is it correct that you only train on a single environment configuration? What happens if the training distribution becomes more variate?

Q3: Fig 3: you show that SAVI omits one agent and so misses that slot. I don't understand why your loss is prohibiting this. Isn't that a classical hyperparameter problem for SAVI?

Q4: Also Fig 3: where are the targets in the MPE test-time picture?

Q5: what is the dimensionality of the latent action space?

Q5: I would like to understand better what really happens, so more analysis could help.
What are the latent actions for passive objects / passively moving objects?

Q6: If there are many agents in a scene and I have access to actions that were performed for one agent (e.g. the Player), can you model be used to find out "who am I?"

Q7: What about a real downstream task, namely ILFO that you prominently mention. What would be the performance of a policy that you train on the inferred actions?

Q8: Section 5.1: you check this for one action. Do you find this for all actions? The result is quite anecdotal. I don't see how it supports the claim that generally shared action representations are found. Also, I find Fig 4 not clear to interpret. The plot does not show the correlation, does it?

Q9: Section 5.2: I am wondering about the training set. It is really just this one scenario, or many paths of the agent to wards the goal with obstacles at different locations along the path?

Comment:
Related work:
https://arxiv.org/abs/2110.06149 also uses the same set of models to create an action informed latent space (however, not factorized).
There are a number of newer works on slot attention for video prediction, such as Singh et al: STEVE and Manasyan et al: SlotContrast

---

### Official Review · Reviewer_EFzB · 2025-10-31

**Soundness:** 2
**Presentation:** 3
**Contribution:** 2
**Rating:** 4
**Confidence:** 3

**Summary:**

This paper presents an variational, slot‑based model that discovers agent‑centric state factors directly from pixels by jointly learning an inverse dynamics (latent action inference), a forward dynamics (state prediction given latent action), and a policy prior over latent actions. It demonstrates the generalizability to novel agents/goals/layouts in Minigrid, Overcooked, and MPE. The learned representation emerges properties such as agent‑invariant action codes and teleological (rational‑action) predictions. VAD’s sits at the intersection of object‑centric video models (Slot Attention; SAVi; SlotFormer) and imitation/representation from observation with latent actions (ILPO; GAIfO). Relative to these lines, its main novelty is factoring the latent‑action inference per slot to discover agents from pixels, rather than assuming agents are known.

**Strengths:**

* The paper’s perspective is well‑motivated by classic agency perception and theory of mind literature and by recent evidence that current VLMs still lag behind humans in intuitive psychology.

* The author introduces a simple but compelling inductive bias: per‑slot transitions as caused by a slot‑specific latent action encourage specialization of “agent slots” vs “environment slots.” The variational formulation enables a unsupervised learning form which reduces data collection burden.

* The three benchmarks (Minigrid goal transfer, Overcooked layout flip, MPE novel agent) target different generalization axes. Reporting linear‑probe action/goal prediction from slots is a standard way to assess whether the right information is present in the representation.

* The teleological test is interesting, and the connection to teleological reasoning in infants is relevant. It demonstrates the possibility of enhancing agency by integrating latent action modeling as intention learning.

**Weaknesses:**

* Object-centric latent action learning framework including IDM and FDM has been established in [1], which shares very similar methodology but not in the framing of agent discovery and application for multi-agent cases. What's the connections and distinctions between the proposed approach and the previous work?

* There are still places for improvement in the experiment setups. The baseline coverage can be improved by considering work from latent-action imitation from observation (e.g., LAPO [2], DIFO [3]), more timely object-centric dynamics modeling (e.g., SlotFormer [4]), as authors have reviewed in the related work.

* Without constraints, the policy may track the inverse posterior rather than an actual policy. The author may consider adding ablations with an isotropic prior (no learned policy), or with action‑agnostic forward dynamics, to show the prior’s distinct contribution.

[1] Klepach, Albina, et al. "Object-Centric Latent Action Learning." arXiv preprint arXiv:2502.09680 (2025).

[2] Schmidt, Dominik, and Minqi Jiang. "Learning to act without actions." arXiv preprint arXiv:2312.10812 (2023).

[3] Huang, Bo-Ruei, et al. "Diffusion imitation from observation." Advances in Neural Information Processing Systems 37 (2024): 137190-137217.

[4] Wu, Ziyi, et al. "Slotformer: Unsupervised visual dynamics simulation with object-centric models." arXiv preprint arXiv:2210.05861 (2022).

**Questions:**

I have left most of my concerns and suggestions in the weakness section. Here are additional questions:

* Could authors consider efficiency‑violation and occlusion variants in the teleological reasoning experiment, to rule out simple shortest‑path heuristics?

* What would happen when the number of agents exceeds the number of slots? Is there graceful degradation or slot reallocation, e.g. only predominant agents are learned in the slots?

---

### Note · Authors · 2025-11-18

I have read and agree with the venue's withdrawal policy on behalf of myself and my co-authors.